# Design and Ductile Behavior of Torsion Configurations in Material Extrusion to Enhance Plasticizing and Melting

**DOI:** 10.3390/polym13183181

**Published:** 2021-09-19

**Authors:** Ranran Jian, Weimin Yang, Mohini Sain, Chuanwei Zhang, Lupeng Wu

**Affiliations:** 1College of Electromechanical Engineering, Qingdao University of Science and Technology, Qingdao 266061, China; jianrr@foxmail.com; 2College of Mechanical and Electrical Engineering, Beijing University of Chemical Technology, Beijing 100029, China; yangwm@mail.buct.edu.cn; 3Centre for Biocomposites and Biomaterial Processing, Department of Mechanical and Industrial Engineering, University of Toronto, Toronto, ON M5S 3B3, Canada; m.sain@utoronto.ca; 4College of Mechanical and Electrical Engineering, Qingdao University, Qingdao 266071, China; zhangchuanwei3722@163.com

**Keywords:** ductile behavior, orthogonal test, torsion configuration, heat transfer, melting capability

## Abstract

In the present work, the ductile formation mechanism of a newly proposed torsion configuration has been investigated. One of the unique attributes of this paper is the first-time disclosure of the design and fabrication of a novel prototype screw with torsional flow character validating the orthogonal test model experimentally. The torsional spiral flow patterns that occurred in the torsion channel cause a ductile deformation of polymer in the form of a spiral, which in turn enhances the radial convection, achieving an effective mass transfer of material from the top region to the bottom region and vice versa. Furthermore, the characteristic parameters of torsion configuration have a significant influence on the plasticizing and melting capability of polymer. By range analysis and weight matrix analysis, the best factor and level combination was obtained. Results indicated that the aspect ratio of the torsion channel is almost equal to 1, and the plasticizing and melting capability of polymer is optimal. This novel design innovation offers a paradigm shift in the energy-efficient plasticization of polymer compounds.

## 1. Introduction

Currently, the issues of energy consumption and carbon emissions have become the focus of international concern, and there is an increasing need for a low-carbon and high-efficiency technology upgrade, especially in the chemical industry [1,2,3]. An extruder is a typical piece of chemical equipment, and its plasticizing and melting performance is an important part of polymer extrusion process control, divided into plasticizing efficiency and plasticizing uniformity, which is accompanied by high energy consumption. In polymer processing, the energy consumed by melting accounts for 70% to 80% of the total input energy. The efficiency of the melting process directly determines the entire process, and the melting process largely determines the stability and quality of the subsequent process. However, due to the phase transition and solid–liquid two-phase transport, the flow and heat transfer process is quite complex. Accordingly, it is very important to study the fluid ductile flow behavior to enhance melting and heat transfer in the melting zone for low-carbon, energy-saving and improving the quality of products. Screw structure optimization is an important research and development direction.

The extrusion process generally involves three processes: solid particle compaction, melt plasticization and melt homogenization. Due to the high viscosity, non-Newtonian and other special physical properties of polymer melt, it is difficult to plasticize uniformly, so the homogenization process needs to be extended by using screws with a large length-diameter ratio to make up for the uneven plasticization, which greatly reduces processing efficiency and increases energy consumption. Wang et al. [4] denoted that improving the processing efficiency and plasticizing uniformity essentially requires effective control of the flow field and temperature field in the extrusion process, such as the introduction of elongational flow in the extrusion system.

In fact, in terms of melting efficiency and power consumption, the elongational flow is far superior to the shear-type flow found in most mixing systems. In this connection, many scholars and researchers have made great efforts. Qu et al. [5] proposed and studied a vane extruder based on the volumetric tensile deformation of polymers. Results indicated that the vane plasticizing and conveying units caused mandatory deformation of particles, resulting in plastic energy dissipation to accelerate the solid particle melting and shorten the thermo-mechanical history. Wu et al. [6] further studied the formation mechanism of the elongational flow field in the vane extruder according to the basic structural and flow patterns of vane plasticizing and conveying units. Wen et al. [7] designed an eccentric rotor extruder based on continuous elongational flow. Results showed that mechanical energy was dissipated into heat via the deformation of particles due to the eccentricity, which increased the melting efficiency and decreased the energy consumption. Rauwendaal et al. [8] developed various screw configurations based on elongational rheology, such as CRD (Chris Rauwendaal Dispersive)mixing section, CRD barrier screw and CRD separation-type screw. The main structural feature of the CRD mixing section is that the flank of screw flights is inclined and conical grooves are set in the screw flights; the main structural feature of the CRD barrier screw is that a slanted part is added to the top of the barrier flights and the main structural feature of the CRD separation-type screw is that a slope is set at the top of the secondary screw flights, all of which causes elongational flow by setting converging wedge channels. The extensional flow created in the wedge-shaped zone reduces the viscous dissipation and power consumption and enhances the stress of the melt, so the melting and mixing are more efficient.

Furthermore, chaos theory is applied to polymer processing by some scholars. For example, Xu et al. [9] developed a chaos screw by employing a reciprocating baffle. Through inserting a periodic barrier element into the screw, the steady flow field is disturbed, and chaotic flow is generated in the channel, thereby improving the efficiency of mixing and melting. Furthermore, Xu et al. [10] studied the influence of geometric ratio and height of baffles on the chaotic mixing and achieved an optimal geometric ratio. Zeng and Qu [11] introduced the vibration to obtain chaos control in the melting process of polymer. Results indicated that the addition of the vibration force field to the melting process can enhance the polymer plasticization. Zeng et al. [12] analyzed the polymer plasticization in the developed electromagnetic dynamic extruder by establishing a mathematical-physical model. The results showed that the processing temperature and energy consumption were reduced, and the plasticizing rate was increased over 60% by adding a vibration force field.

A significant amount of experience has been gained over decades of active research on rheology and energy dissipation to accelerate the melting rate of polymers. However, heat transfer through a barrel is also an important heat resource to melt the polymer, and there are only a few reports on the heat transfer enhancement in the screw extrusion. Due to the poor thermal conductivity of polymer, the convective heat transfer effect inside the melt and between the polymer and external heater is insufficient, so improving the convective heat transfer is also an important direction to accelerate melting. Rauwendaal [13] simulated the melt flow and temperature evolution in a single screw extruder (SSE) by using the finite element method (FEM), considering the influence of geometry parameters of screws. Furthermore, Rauwendaal [14] developed an analytical theory to predicate the melt temperature profiles in the extrusion process by analyzing the flow and heat transfer behavior of both single and twin screw extruders. Teixeira et al. [15] presented a plasticizing model for co-rotating the twin screw extruder to predicate the evolution of temperature and mechanical power consumption. All of the above researchers made a preliminary analysis of heat transfer; however, their focus is on temperature distribution without considering heat transfer efficiency. In order to improve the cooling capacity in polystyrene foam extrusion, Rauwendaal [16] fabricated a high heat transfer (HHT) screw and achieved effective mass transfer from the center region to the outside region in the screw channel. In our previous work, Jian et al. [17,18] developed a torsion extrusion technology based on multi-field synergy, inducing torsional spiral flow in the screw channel to improve phase-to-phase molecular and thermal mobility, achieving effective mass transfer from the top region to the bottom region in the screw channel. Then, Jian et al. [19,20] studied the heat transfer and mixing capability of a torsion screw compared with three typical geometric forms commonly employed in practical application: conventional screw, Maddock screw and Pin screw. The result indicated that the torsion screw by setting twisted channels displayed superior mixing and heat transfer properties through the self-twisting mass transfer of particles, avoiding over-shear and overheating. This multi-field synergy theory of polymer focuses on the heat transfer enhancement through the barrel and the mass and thermal transfer efficiency inside of the polymer itself, especially in the radial direction, offering a novel perspective to optimize and design the screw.

The torsion configuration is designed to achieve an effective exchange of material from the top region to the bottom region and vice versa, thereby improving the heat transfer and melting efficiency. Considering the torsion configuration has good heat transfer and melting capability, it is necessary to further investigate the influence of structural parameters on performance. Therefore, the authors conducted research on the plasticizing and melting capability by changing parameters of torsion configuration, including the number of flights, the width of flight, and the height of the channel based on the orthogonal test, ascertaining complex ductile deformation behavior and plasticizing properties in the solid–melt transition process.

## 2. Computational Methodology

### 2.1. Physical Model

The three-dimensional model of the newly designed torsion configurations is shown in Figure 1. As shown in Figure 1a, it has several torsion flights along the circumferential direction to divide the screw into several torsion channels. There are two surfaces twisted gradually from 0° to 90° in each two adjacent torsion flights, called 90° twisted surfaces, thereby leading to a torsion spiral flow in the torsion channels. Figure 1b shows the characteristic parameters of the torsion configuration, such as diameter (*D*), root diameter (*d*), number of flights, width of flight (*e*), length of element (*L*) and length of transition (*b*). The torsion channels and flights are evenly divided according to the circle angle *α* and *β*, which determined the width of each channel and flight when the screw diameter is fixed, in turn, the number of flights is determined by the sum of angle *α* and *β*. The height of the torsion channel is defined by half of the difference between the screw diameter (*D*) and screw root diameter (*d*).

Here, we focus on the melting section of the screw to study the ductile behavior in the phase transformation process of solid–melt, and the experimental screw is shown in Figure 2. There is a set of torsion configurations in the screw, each with three, all of which are located close to the outlet side by side. There is a screw zone with one pitch before the torsion configurations and a polished rod area after the torsion ones, avoiding the influence of the inlet and outlet boundary on the results. Table 1 shows the basic parameters of the screw in the simulation.

### 2.2. Governing Equations and Mesh System

The plasticizing and melting of the polymer in the screw channel is a process related to heat and mass transfer, multiphase flow, and complicated rheology and ductile behavior. In this case, the model fluid was polypropylene (PP), assumed as laminar flow in a non-isothermal steady state, and was incompressible. The non-slip condition was employed at the boundary. Inertial force can be neglected compared with viscous force caused by the high viscosity of PP. Based on these assumptions, the specific governing equations are as follows:

The continuity equation
(1)$$\frac{\partial u_{i}}{\partial x_{i}}\;=\;0$$

The momentum equation
(2)$$\frac{\partial P}{\partial x_{i}}\;=\;\frac{\partial }{\partial x_{j}}\left(\eta \frac{\partial u_{i}}{\partial x_{j}}\right)$$

The energy equation
(3)$$\rho C_{p}\left(u_{i}\frac{\partial T}{\partial x_{i}}\right)\;=\;\lambda \frac{\partial^{2}T}{\partial x_{i}^{2}}\;+\;\varphi$$

Bird-Carreau and Arrhenius models were adopted to describe the apparent viscosity of PP (Equation (4)), involving the effects of temperature and shear, which is compatible with most polymers. The material properties of the selected polymer determining the parameters in the constitutive equation (Equation (4)) are shown in Table 2.
(4)$$\eta \;=\;\left[\eta_{\infty }\;+\;\left(\eta_{0}\;-\;\eta_{\infty }\right){\left(1\;+\;t_{0}^{2}{\dot{\gamma }}^{2}\right)}^{\frac{n\;-\;1}{2}}\right]exp\left[\alpha \left(\frac{1}{T\;-\;T_{0}}\;-\;\frac{1}{T_{\alpha }\;-\;T_{0}}\right)\right]$$

In the simulation of such a screw plasticizing process, the fluid was in the state of multiphase flow, experiencing melting from solid to melt, hence the “Solidification and Melting” calculation model was used, and the “Enthalpy-porosity” technology was adopted to assess the ratio of melt grids in the whole flow domain. The ANSYS Fluent 17.0 packaged software (ANSYS, Inc., Canonsburg, PA, USA) was used for the numerical simulation. The mesh models were generated by employing ANSYS ICEM 17.0 (ANSYS, Inc.), and grid independent test was performed to ensure the reliability of the simulation results. As shown in Figure 3, the data hardly change and tend to be smooth and steady when the mesh number is over 1,800,000; thereby, the mesh systems used in the present work were around 2,000,000 cells.

### 2.3. Boundary Conditions

The simulation models were the fluid domain from the barrel wall to the screw root. The flow and thermal boundary conditions employed in the present work are shown in Table 3.

### 2.4. Characterization

The plasticizing efficiency and uniformity are two important aspects to reflect the plasticizing and melting capability. In the numerical simulation, the liquid mass fraction (*LMF*) was used to simulate the proportion of melt meshes in the solid–melt two-phase flow field, which can reveal the plasticizing efficiency in the extrusion process. The Nusselt number (*Nu*) was used to characterize the heat transfer efficiency, and the shear stress (*τ*) was used to reflect the viscous dissipation. The standard deviation of temperature (*T_SD_*) at the outlet was used to characterize the plasticizing uniformity, which can be expressed as:(5)$$\delta \left(T\right)\;=\;\sqrt{\frac{\sum {\left(T_{i}\;-\;\bar{T}\right)}^{2}\;\cdot \;\Delta A_{i}}{A}}$$
where *T_i_* and *A_i_* are the temperature and area of the *i*th cell, *A* is the area of the whole domain, and $\bar{T}$ is the average temperature of the whole domain.

### 2.5. Orthogonal Experiment Design

In this work, three factors of torsion configuration play important roles in determining the plasticizing and melting capability in the screw extrusion process, which are the number of flights, the width of flight and the height of the channel.

As shown in Table 4, Factor A is the number of flights, Factor B is the width of the flight, Factor C is the height of the channel, respectively, in the design of the orthogonal test. The three aforementioned factors and four levels were used to optimize orthogonal experiments, and the design scheme of the orthogonal test, a total of 16 groups, is shown in Table 5. According to the orthogonal test table L16(4^5^), 16 torsion configurations were designed, as shown in Figure 4.

### 2.6. Experimental

In order to confirm the results of the orthogonal experiment in the numerical simulation, an extrusion test was carried out with polypropylene (PP, T30S), provided by the Sinopec Zhenhai Refining and Chemical Company (ZRCC, Ningbo, Zhengjiang, China). The extrusion process of PP was investigated by using a filament die and a modular single screw with a length-diameter ratio of 28:1. All experiments were performed at a constant processing condition, that is, the barrel temperature was set to 200 °C and the die temperature was set to 195 °C. The torsion configurations were arranged at the end of the melting section.

In order to determine the standard deviation of temperature at the outlet, six melt temperature sensors were placed through the holes evenly distributed around the die, and their probes were immersed at various radial locations of the melt, as shown in Figure 5.

## 3. Results and Discussion

### 3.1. The Ductile Formation Mechanism of The Torsion Configuration

A good flow field is essential to obtain a proper ductile behavior of polymer for a screw plasticizing process. Figure 6 illustrates the streamline traces of polymer in the screw channel for case 6. From the flow patterns in the axial direction, as shown in Figure 6a, a torsion-spiral flow occurred in the torsion channel at the position of torsion configurations, inducing a ductile deformation of polymer in the form of a spiral, and thereby, the radial convections were enhanced. From the streamline distribution in Figure 6b, obvious vortices can be found in the vertical cross-section. The reason is that the high viscosity polymer flowing over the torsion channel will generate a torsional rotation under the combined action of the viscous friction on the barrel surface and the diversion by the two 90° twisted surfaces. Similarly, the same phenomenon was observed for other cases. In this way, an effective exchange of material (e.g., mass and heat) from the top region to the bottom region in the screw channel was achieved and vice versa.

Figure 7 shows the liquid mass fraction (*LMF*) distributions near the screw root for both cases 1 and 10. It is obvious that the results of the liquid mass fraction distributions were different when the characteristic parameters of the torsion configuration changed. For case 1, when the polymer was flowing into the second channel of torsion configurations, the values of *LMF* were almost close to 100%. However, the values of *LMF* were not close to 100% until the polymer was flowing to the third channel for case 10. Accordingly, it is necessary and significant to optimize the characteristic parameters of the torsion configuration for obtaining good plasticizing and melting capabilities of polymer.

### 3.2. Analysis of Orthogonal Test Results

According to the orthogonal analysis, the effect of three factors including the number of flights, the width of flight, and the height of the channel, on the liquid mass fraction (*LMF*), the Nusselt number (*Nu*), the shear stress (*τ*), and the standard deviation of temperature (*T_SD_*) at the outlet were studied. The orthogonal experiment result table L16(4^5^) involves three factors, with four levels of each factor, and the test indexes of the results are the liquid mass fraction (*LMF*), the Nusselt number (*Nu*), the shear stress (*τ*), and the standard deviation of temperature (*T_SD_*) at the outlet when the screw speed was 75 r/min, as shown in Table 6. All the indexes of the test results were averaged.

The results of the range analysis can be found in Table 7, Table 8, Table 9 and Table 10. Specifically, *K_ij_* (*i* = A, B, C; *j* = 1, 2, 3, 4) reflects the average of all the result indicators related to the level *j*; the range (*R_i_*) is the difference between minimum and maximum of *K_ij_* for the factor *i* in the same column, representing the importance degree of factor *i*.

#### 3.2.1. Liquid Mass Fraction

Table 7 shows the range analysis results for the liquid mass fraction. In practice, the larger the liquid mass fraction, the better the plasticizing efficiency in the process, that is, the larger value of *K_ij_* is optimum. Thus, the best group is A_1_B_1_C_1_, corresponding to case 1, that is, when the number of flights is eight, the width of flight is 1 mm, and the height of the channel is 3 mm, the average liquid mass fraction of the whole domain is the maximum, which is 79.7%.

As shown in Table 7, the range of each factor is *R_C_* > *R_B_* > *R_A_*. Accordingly, the influence degree of three factors on the liquid mass fraction at 75 r/min is: the height of the channel > the width of flight > the number of flights. Thus, we can find that, in three factors affecting the liquid mass fraction, the height of the channel is the main factor, and the width of flight and the number of flights are second, respectively.

#### 3.2.2. Nusselt Number

Table 8 shows the range analysis results for the Nusselt number. In practice, the larger the Nusselt number, the better the heat transfer in the process, that is, the larger value of *K_ij_* is optimum. Thus, the best group is A_1_B_1_C_1_, corresponding to case 1, that is, when the number of flights is eight, the width of flight is 1 mm, and the height of the channel is 3 mm, the average Nusselt number of the whole domain is the maximum, which is 342.2.

As shown in Table 8, the range of each factor is *R_C_* > *R_A_* > *R_B_*. Accordingly, the influence degree of three factors on the Nusselt number at 75 r/min is: the height of the channel > the number of flights > the width of flight. Thus, we found that, in three factors affecting the Nusselt number, the height of the channel is the main factor, and the number of flights and the width of the flight are second, respectively.

#### 3.2.3. Shear Stress

Table 9 shows the range analysis results for shear stress. Considering that a larger shear stress results in a larger viscous dissipation to accelerate the melting process, that is, the larger value of *K_ij_* is optimum, special attention should be paid to the fact that the shear stress should not be too large to avoid product defects caused by overheating and over-shear; fortunately, this condition did not occur in this orthogonal experiment.

For the purpose of improving the plasticizing and melting, the best group is A_3_B_2_C_4_, corresponding to case 10, that is, when the number of flights is 12, the width of flight is 2 mm, and the height of the channel is 1.5 mm, the average shear stress of the whole domain is the maximum, which is 3.855 kPa.

As shown in Table 9, the range of each factor is *R_C_* > *R_B_* > *R_A_*. Accordingly, the influence degree of three factors on the shear stress at 75 r/min is: the height of the channel > the width of flight > the number of flights. Thus, we can conclude that, in three factors affecting the shear stress, the height of the channel is the main factor, and the width of flight and the number of flights are second, respectively.

Compared with the range analysis results in Table 7, Table 8 and Table 9, it can be seen that the ranges of factor A and factor B for all these three indexes (*LMF*, *Nu*, *τ*) are close in value and much smaller than the ranges of factor C. The evidence indicates that among the three factors, the height of the channel has considerable effect on the liquid mass fraction (*LMF*), the Nusselt number (*Nu*), and the shear stress (*τ*), whereas the width of flight and the number of flights have little effect on these three indexes.

As is well-known, viscous dissipation and convective heat transfer are two main sources of polymer temperature rise. From Table 7 and Table 8, it can be found that the trend of the liquid mass fraction and the Nusselt number is identical. However, the law of the range values between shear stress and liquid mass fraction is not obvious, compared with the range analysis results in Table 7 and Table 9. The possible explanation is the torsion configurations can hardly affect the shear behavior due to a lack of strong shear structures; however, the torsion configurations have significant influence on the Nusselt number and heat transfer, which is also what it was designed for. Specifically, the heat transfer process is very important for plasticizing and melting capability in the extrusion.

#### 3.2.4. Standard Deviation of Temperature

Table 10 shows the range analysis results for the standard deviation of temperature at the outlet. In practice, the smaller the standard deviation of temperature, the better the plasticizing uniformity in the process, that is, the smaller value of *K_ij_* is optimum. Thus, the best group is A_4_B_3_C_4_, which was obviously not in this orthogonal experiment scheme, reflecting that the orthogonal analysis can find the best combination, even the unplanned experiment group. According to the results of the orthogonal experiment, the best group of this test is A_4_B_3_C_2_, corresponding to case 15, that is, when the number of flights is 14, the width of the flight is 3 mm, and the height of the channel is 2.5 mm, the average standard deviation of temperature at the outlet is the minimum, which is 0.212 °C.

As shown in Table 10, the range of each factor is *R_A_* > *R_B_* > *R_C_*. Accordingly, the influence degree of three factors on the standard deviation of temperature at 75 r/min is: the number of flights > the width of flight > the height of the channel.

Specifically, the ranges of factor A and factor B are very close, and the combination of them determines the width of the torsion channel. Moreover, factor C has little influence on the standard deviation of temperature because of its low range value. In order to further validate the influence of the torsion configurations on the standard deviation of temperature, we defined the aspect ratio of the torsion channel as the ratio of width to height and simulated the temperature distributions in a single torsion channel. Table 11 shows the dimensions of the four single torsion channels employed in the simulation, and we kept the height constant owing to its low influence. 

Figure 8 shows the temperature distributions of different axial cross-section in the single torsion channels. It is obvious that the polymer fluid with high temperature near the upper barrel surface went down to the screw surface due to the torsional rotation flow for all these four cases, thereby the radial transfer and uniform distribution of thermal energy were realized. Moreover, the standard deviation of temperature at the outlet was calculated, as shown in Figure 8. Results indicated that the standard deviation of the temperature at the outlet reduces first and then improves with the increasing aspect ratio of the torsion channel, and there is an intermediate value, 11.87 °C in the simulation, when the aspect ratio was equal to 1. This is consistent with the range analysis results for the standard deviation of the temperature at the outlet, that is, the aspect ratio for case 15 was also close to 1. Accordingly, the molten and mass transfer model in one torsion channel can be established, as shown in Figure 9. An effective mass and heat transfer of material is achieved from the top region to the bottom region by torsional flow in the torsion channel and vice versa.

Above all, the optimal combination of factors and levels for the liquid mass fraction (*LMF*) and the Nusselt number (*Nu*) is A_1_B_1_C_1_, whereas the optimal combination is A_3_B_2_C_4_ for the shear stress (*τ*) and is A_4_B_3_C_4_ for the standard deviation of temperature (*T_SD_*), respectively. In a word, the influence of characteristic parameters of the torsion configuration on different indexes is different. Therefore, we should consider each index comprehensively, weigh the requirements to maximize the benefits of the whole life cycle in the extrusion, including efficiency, energy consumption and quality, and finally get the optimal combination.

### 3.3. Screw Speed Analysis

To display the influence law of the factors and levels on the test indexes and confirm the effect of screw speeds on the results, Figure 10 illustrates intuitively the correlation between the factors and the test indexes at different screw speeds. It can be found from Figure 10a,b, the variation trend of the impact of different factors and levels on the liquid mass fraction (*LMF*) and the standard deviation of temperature (*T_SD_*) at various screw speeds were similar; in particular, the similar phenomenon is more obvious at high screw speeds for the standard deviation of temperature (*T_SD_*). This shows from a certain side that the above orthogonal analysis is feasible to select a certain speed, 75 r/min in the analysis.

As we know from the range analysis results for liquid mass fraction, factor C, the height of the channel, is the main factor affecting the liquid mass fraction. From Figure 10a, it can be found that the liquid mass fraction reduces markedly with the decrease in the height of the channel. Likewise, from the range analysis results for the standard deviation of temperature, factors A and B, the number of flights and the width of flight, are the main factors that affect the standard deviation of temperature. From Figure 10b, the standard deviation of temperature at the outlet shows a downward trend with increasing the number of flights and the width of the flight.

### 3.4. Weight Matrix Analysis

By weight matrix analysis, the specific weight of each factor level can be quantitatively obtained, and the result is illustrated in Table 12. The weight calculation process is relatively fixed, and Yao et al. [21] described the weight calculation formulas in detail in the relevant literature; thus, they will not be repeated here.

According to the calculation results, the influence weights of different factors for liquid mass fraction (*LMF*) are A = 0.1413, B = 0.1930, C = 0.6657, the impact order of three factors for *LMF* is: C > B > A, the weight value of factor C is the largest and much larger than those of factors A and B. Similarly, the influence weights of different factors for the standard deviation of temperature (*T_SD_*) are A = 0.4348, B = 0.4131, C = 0.1521, the impact order of three factors for *T_SD_* is: A > B > C, the weights of factor A and factor B are close and much larger than that of factor C. The results of the weight matrix analysis and the range analysis are in good agreement. Considering the comprehensive effect of the three factors on the liquid mass fraction (*LMF*) and standard deviation of temperature (*T_SD_*) at the outlet, the best group is A_4_B_3_C_1_, and the worst combination is A_1_B_1_C_4_.

### 3.5. Experimental Validation

In order to validate the orthogonal analysis results, the optimal (A_4_B_3_C_1_, marked with A*) and the worst (A_1_B_1_C_4_, marked with B*) torsion configurations were designed and fabricated to run an extrusion experiment, and as a control, the conventional screw (marked with C*) was also employed in the experiment.

By measuring the melt temperature before and after the position of torsion configurations in the melting section, we can obtain the axial temperature distributions, as shown in Figure 11. It is obvious that the temperatures in position 2 were different after the polymer flowed through the torsion configurations, that is, the temperature for screw A* was the highest, the temperature for screw B* was second, and the temperature for screw C* was the lowest, respectively, indicating screw A* had the best melting capability. Therefore, the order of the melting capability for the three screws at 75 r/min is: A* > B* > C*, which is consistent with the orthogonal results.

Figure 12 shows the temperature distributions in the extruder die. As shown in Figure 12a, the trend of the radial temperature profiles was the same for all of the testing screws, that is, the temperature decreases from the center to the wall of the extrusion die. However, the maximum difference of temperature was different, and the order of the temperature difference for the three screws at 75 r/min is: A* < B* < C*. The standard deviation of temperature at different screw speeds was calculated and is shown in Figure 12b. We can see that the standard deviation of temperature increases with the increase in the screw speed, but among all the screw speeds, screw A* had the smallest standard deviation, followed by screw B*, and screw C* had the largest standard deviation, respectively. In fact, the plasticizing and melting capability of screw A* are better than that of screw B*, which is consistent with the orthogonal analysis results.

## 4. Conclusions

In the study, ductile behavior of torsion configurations was numerically studied by adopting CFD technology and orthogonal analysis. The velocity and temperature distribution inside the screw channel was recorded, and the ductile formation mechanism of the torsion configuration was discussed. The main factors affecting the plasticizing and melting capability of torsion configurations and the optimal group of factors and levels were obtained by range analysis, together with the weight matrix analysis of the orthogonal test. The specific conclusions are as follows:

(1) From the flow and thermal conditions inside the screw channel, we found that the torsion spiral flow patterns occur in the torsion channel, inducing a ductile deformation of polymer in the form of a spiral, which in turn enhances the radial convection, resulting in a good plasticizing and melting capability. In addition, the characteristic parameters of torsion configuration have a significant influence on the plasticizing and melting capability of polymer.

(2) The range analysis of the orthogonal experiment shows that the orders of the influence degree of factors on the liquid mass fraction (*LMF*), the Nusselt number (*Nu*) and the shear stress (*τ*) are: the height of the channel > the width of flight ≒ the number of flights (C > B ≒ A), whereas the order of the standard deviation of temperature (*T_SD_*) is: the number of flights ≒ the width of flight > the height of the channel (A ≒ B > C), respectively. The results indicated that the influence of characteristic parameters of torsion configuration on different indicators is different, which should be considered comprehensively.

(3) The weight matrix method of the orthogonal experiment shows that the influence weights of different factors on the liquid mass fraction (*LMF*) are A = 0.1413, B = 0.1930, C = 0.6657, whereas the weights for the standard deviation of temperature (*T_SD_*) are A = 0.4348, B = 0.4131, C = 0.1521. Results indicated the height of the channel is the main factor to determine the liquid mass fraction (*LMF*), and combinations of the number of flights and the width of flight, reflecting the width of the torsion channel, are the main factors to jointly determine the standard deviation of temperature (*T_SD_*).

(4) Combining the range analysis and weight matrix analysis, we found that the optimal combination is A_4_B_3_C_1_; in other words, when the number of flights is 14, the width of flight is 3 mm, and the height of the channel is 3mm, i.e., the aspect ratio of the torsion channel is almost equal to 1, the plasticizing and melting capability of the polymer is the best, which was validated by the extrusion experiment in practice. Thereby, it can offer a reference for the design and optimization of torsion configurations and provide an example for energy-efficient plasticization of polymers.

(5) Although the torsion configuration performs good radial convection and heat transfer enhancement, it does not have positive displacement transport characteristics compared with a helically grooved screw because its torsion flights are parallel to the screw axis. By redesigning the torsion flights to the helical type, it is expected to improve the positive displacement transport of the torsion configuration, which is one of our ongoing studies.

## Figures and Tables

**Figure 1 polymers-13-03181-f001:**
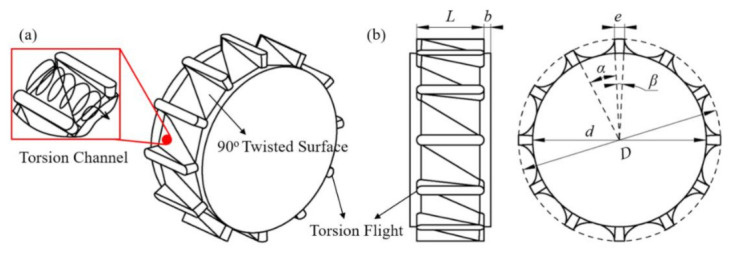
Three-dimensional model (**a**) and the characteristic dimensions (**b**) of the torsion configuration.

**Figure 2 polymers-13-03181-f002:**
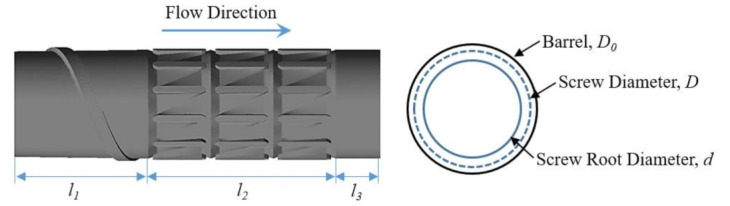
The simulation object-torsion screw.

**Figure 3 polymers-13-03181-f003:**
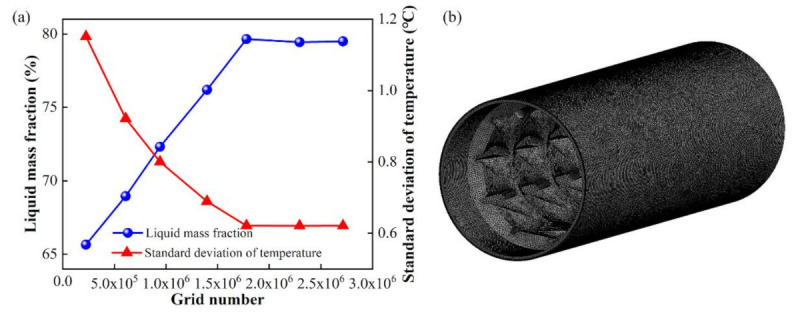
Grid independent test for data of liquid mass fraction and standard deviation of temperature (**a**), and the discrete model with about 2,100,000 cells (**b**).

**Figure 4 polymers-13-03181-f004:**
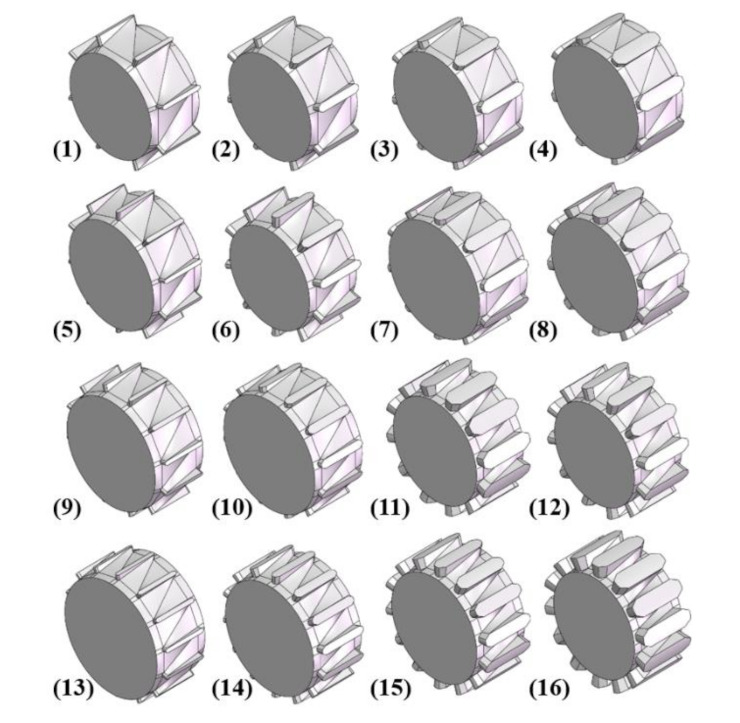
The torsion configurations in each test group among the total 16 groups.

**Figure 5 polymers-13-03181-f005:**
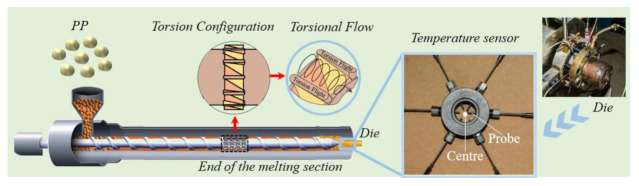
The experimental extruder with a radial temperature measurement in the position of the die.

**Figure 6 polymers-13-03181-f006:**
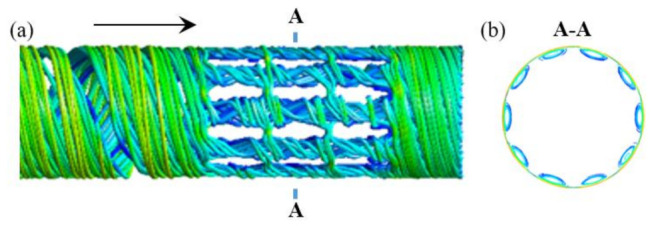
Streamline distributions of the axial direction (**a**) and the vertical cross-section (**b**) in the screw channel for case 6.

**Figure 7 polymers-13-03181-f007:**
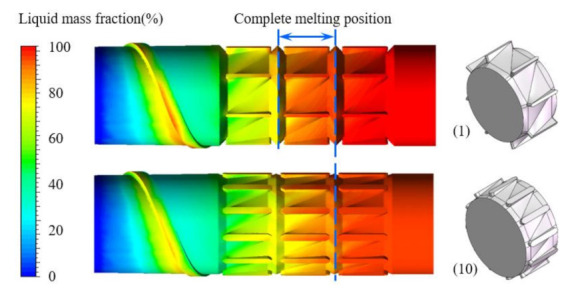
Liquid mass fraction distributions near the screw root for cases 1 and 10.

**Figure 8 polymers-13-03181-f008:**
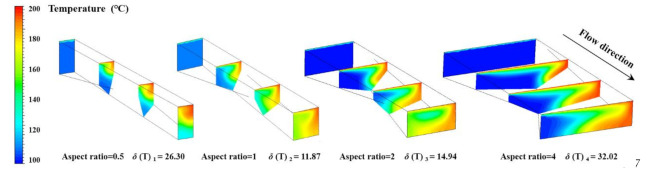
Temperature distributions in the single torsion channels.

**Figure 9 polymers-13-03181-f009:**
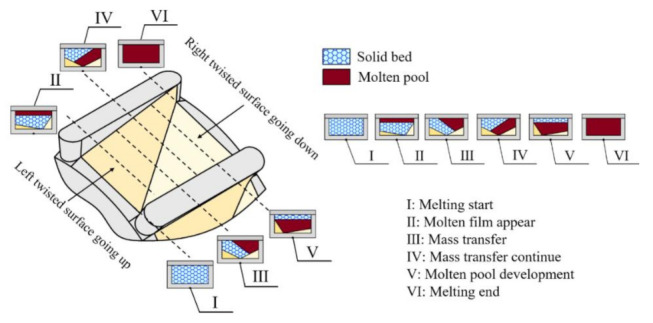
The molten and mass transfer mechanism model in one torsion channel.

**Figure 10 polymers-13-03181-f010:**
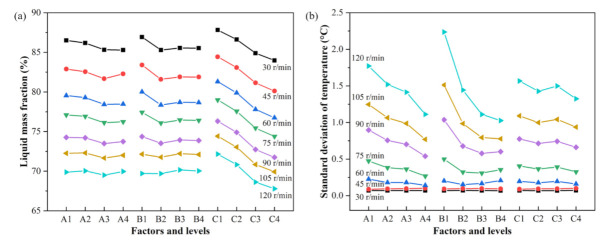
Variation curve of *LMF* (**a**) and *T_SD_* (**b**) with factors and levels at different screw speeds.

**Figure 11 polymers-13-03181-f011:**
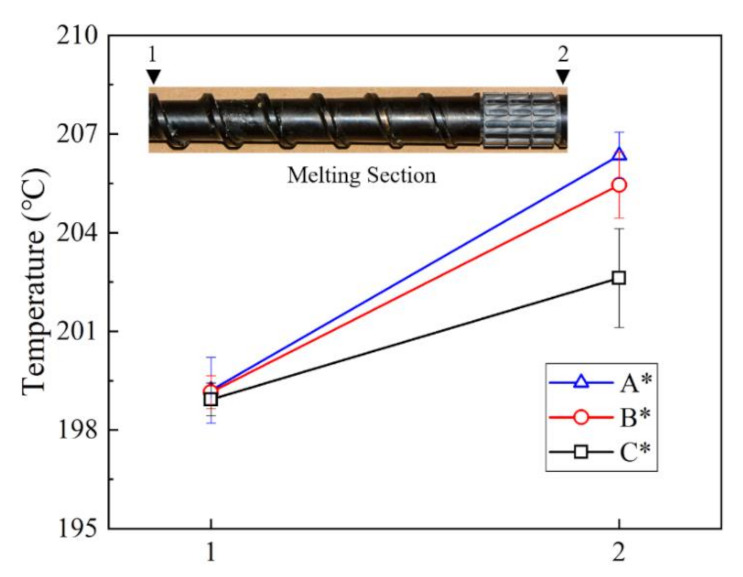
Axial temperature profile for three screws at 75 r/min.

**Figure 12 polymers-13-03181-f012:**
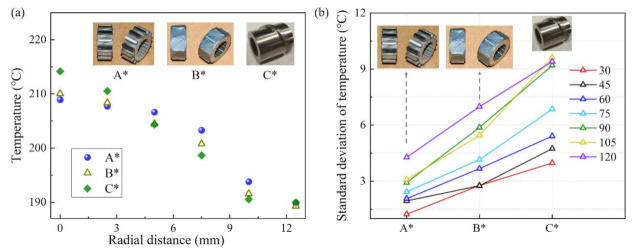
Temperature profile in the extrusion die: (**a**) radial temperature distribution at 75 r/min; (**b**) standard deviation of temperature at different screw speeds.

**Table 1 polymers-13-03181-t001:** Basic physical parameters of the screw.

Dimension	Symbol	Value/mm
Screw diameter	*D*	30
Barrel diameter	*D* _0_	30.4
Length of screw zone	*l* _1_	30
Length of torsion configuration	*L* + 2*b*	14
Length of torsion zone	*l* _2_	42
Length of polished rod area	*l* _3_	10

**Table 2 polymers-13-03181-t002:** Material properties of polypropylene.

Parameters	Symbol	Value
Density	*ρ*	910 kg/m^3^
Thermal conductivity	*λ*	0.2 W/(m·°C)
Specific heat capacity	*C_p_*	2300 J/(kg·°C)
Pure solvent melting heat	∆*H_f_*	150,000 J/kg
Solidus Temperature	*T_S_*	100 °C
Liquidus temperature	*T_L_*	170 °C
Zero shear viscosity	*η* _0_	9650 Pa·s
Viscosity at an infinite shear rate	*η_∞_*	0 Pa·s
Non-Newtonian index	*n*	0.48
Natural time	*t* _0_	0.3664 s
Coefficient of temperature sensibility	*α*	2000 °C^−1^
Reference temperature	*T_α_*	0 °C
Absolute zero	*T* _0_	−273.15 °C

**Table 3 polymers-13-03181-t003:** Boundary conditions.

Screw Model	Boundary	Flow Conditions	Thermal Conditions
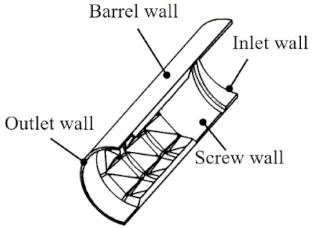	Inlet wall	Velocity inflow ^1^	100 °C
	Outlet wall	Pressure = 3 MPa	185 °C
	Barrel wall	30, 45, 60, 75, 90, 105, 120 r/min	Heat flux, Q = 40,000W/m^2^
	Screw wall	Stationary surface	Insulated surface

^1^ Velocity varied with the screw speed.

**Table 4 polymers-13-03181-t004:** Factors and levels of the orthogonal experiment.

**Level**	**Factor A**	**Factor B**	**Factor C**
1	8	1 mm	3.0 mm
2	10	2 mm	2.5 mm
3	12	3 mm	2.0 mm
4	14	4 mm	1.5 mm

**Table 5 polymers-13-03181-t005:** Test groups of the orthogonal experiment.

**Case**	**Factor A**	**Factor B**	**Factor C**
1	8	1 mm	3.0 mm
2	8	2 mm	2.5 mm
3	8	3 mm	2.0 mm
4	8	4 mm	1.5 mm
5	10	1 mm	2.5 mm
6	10	2 mm	3.0 mm
7	10	3 mm	1.5 mm
8	10	4 mm	2.0 mm
9	12	1 mm	2.0 mm
10	12	2 mm	1.5 mm
11	12	3 mm	3.0 mm
12	12	4 mm	2.5 mm
13	14	1 mm	1.5 mm
14	14	2 mm	2.0 mm
15	14	3 mm	2.5 mm
16	14	4 mm	3.0 mm

**Table 6 polymers-13-03181-t006:** Orthogonal experimental schemes and results at 75 r/min.

Case	Factor A	Factor B	Factor C	*LMF*/%	*Nu*	*τ*/kPa	*T_SD_*/ °C
1	1	1	1	79.70	342.2	2.504	0.625
2	1	2	2	78.75	315.9	2.832	0.418
3	1	3	3	76.38	274.5	3.331	0.439
4	1	4	4	73.58	240.8	3.806	0.410
5	2	1	2	78.95	318.8	2.765	0.499
6	2	2	1	78.01	308.9	2.942	0.381
7	2	3	4	74.03	242.7	3.804	0.290
8	2	4	3	76.68	277.1	3.241	0.360
9	3	1	3	75.67	266.7	3.363	0.550
10	3	2	4	74.56	246.8	3.855	0.276
11	3	3	1	78.57	317.4	2.871	0.294
12	3	4	2	75.68	271.4	3.382	0.329
13	4	1	4	75.36	254.6	3.453	0.327
14	4	2	3	73.02	242.2	3.794	0.214
15	4	3	2	76.87	288.9	3.120	0.212
16	4	4	1	79.69	339.2	2.640	0.319

**Table 7 polymers-13-03181-t007:** Range analysis results for liquid mass fraction.

Parameters	Factor A	Factor B	Factor C
*K_i_* _1_	77.10	77.42	78.99
*K_i_* _2_	76.92	76.09	77.56
*K_i_* _3_	76.12	76.46	75.44
*K_i_* _4_	76.24	76.41	74.38
*R_i_*	0.98	1.33	4.61

**Table 8 polymers-13-03181-t008:** Range analysis results for Nusselt number.

Parameters	Factor A	Factor B	Factor C
*K_i_* _1_	293.35	295.58	326.93
*K_i_* _2_	286.88	278.45	298.75
*K_i_* _3_	275.58	280.88	265.16
*K_i_* _4_	281.23	282.13	246.23
*R_i_*	17.77	17.13	80.70

**Table 9 polymers-13-03181-t009:** Range analysis results for shear stress.

Parameters	Factor A	Factor B	Factor C
*K_i_* _1_	3.12	3.02	2.74
*K_i_* _2_	3.19	3.36	3.02
*K_i_* _3_	3.37	3.28	3.43
*K_i_* _4_	3.25	3.27	3.73
*R_i_*	0.25	0.34	0.99

**Table 10 polymers-13-03181-t010:** Range analysis results for the standard deviation of temperature.

Parameters	Factor A	Factor B	Factor C
*K_i_* _1_	0.47	0.50	0.40
*K_i_* _2_	0.38	0.32	0.36
*K_i_* _3_	0.36	0.31	0.39
*K_i_* _4_	0.27	0.35	0.33
*R_i_*	0.20	0.19	0.07

**Table 11 polymers-13-03181-t011:** Dimensions of the single torsion channel.

Case	Length	Height	Width	Aspect Ratio
I	15 mm	2 mm	1 mm	0.5
II	15 mm	2 mm	2 mm	1
III	15 mm	2 mm	4 mm	2
IV	15 mm	2 mm	8 mm	4

**Table 12 polymers-13-03181-t012:** Weight matrix analysis schemes and results at 75 r/min.

Parameters	A1	A2	A3	A4	B1	B2	B3	B4	C1	C2	C3	C4
*LMF*/%	0.03556	0.03548	0.03511	0.03516	0.04877	0.04793	0.04817	0.04814	0.17163	0.16853	0.16391	0.16162
*T_SD_*/ °C	0.08192	0.10127	0.10700	0.14460	0.07399	0.11484	0.11982	0.10440	0.03466	0.03851	0.03593	0.04308

## Data Availability

The data presented in this study are available on request from the corresponding author. The data are not publicly available at this time as the data also forms part of an ongoing study.

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
