# Peer review of "Design and Ductile Behavior of Torsion Configurations in Material Extrusion to Enhance Plasticizing and Melting"

_polymers, 2021, doi:10.3390/polym13183181_

Round 1
Reviewer 1 Report
Dear Authors,
The paper presents interesting results of the optimization using Taguchi methods coupled with FEM. This paper is a good example how to properly design an "experiment", but in fact I would like to propose few comments in order to improve manuscript quality:
- Section nomenclature is highly recommended to extend with list of the used symbols and abbreviations
- Section 2.2. should be coupled with 2.3. Pleae also show the discrete model from ANSYS.
- It is not fully clear how was validated FEM analysis? Did Authors check any results from FEM using experimental analysis? Can you show any sensors used in real experimental-set-up?
- Line 356 - it is reported results „11.873 °C” – How it was obtained with this accuracy up to 0.001?
Best regards
Rev
Author Response
Please see the attachment, thank you!

Reviewer 2 Report
In this paper Authors investigated the ductile formation mechanism of a newly proposed torsion configurations. The Authors presented design and fabrication of a novel prototype screw with torsional flow character validating the orthogonal test model experimentally. The Authors say that by range analysis and weight matrix analysis, the best factor and level combination was obtained. Below I presented some remarks that came to my mind during reading:
- In my opinion the Introduction can be improved. The Authors should state more precisely what is special, unexpected, or different in their approach and identify a precise gap in the current state of knowledge that needs to be filled - a gap that is being addressed by their research.
- Unfortunately, the presented paper lacks confrontation of the obtained results with other research. From my point of view it has no justification not conducting a comparison with similar works already published. Such comparison significantly raises the meaning of the presented paper. Could the Authors of the paper compare the proposed method with other methods used in the literature for the same purpose?
- In the Conclusions the Authors should present the findings also highlighting current limitations of their study, and briefly mention some precise directions that they intend to follow in their future research work.
- References should be prepared in accordance with the Energies template.
Author Response
Please see the attachment, thank you!
